# Inverted U-shaped association between bacillary dysentery and temperature: A new finding using a novel two-stage strategy in multi-region studies

Wei Wang[1,2,3]☯, Yunqiong Wang[1,2]☯, Lin Chen[3], Bo Zhou[1,2]*, Fang Liao[1,2]*

**1** Sichuan Provincial Center for Mental Health, Sichuan Provincial People's Hospital, School of Medicine, University of Electronic Science and Technology of China, Chengdu, China, **2** Key Laboratory of psychosomatic medicine, Chinese Academy of Medical Sciences, Chengdu, China, **3** West China School of Public Health and West China Fourth Hospital, Sichuan University, Chengdu, China

☯ These authors contributed equally to this work.
* tonyac7721@163.com (BZ); realseo@126.com (FL)

**Data Availability Statement:** The bacillary dysentery case data can be retrieved from the Chinese Center for Disease Control and Prevention via http://en.chinacdc.cn/. The results can be

## Abstract

### Background

Bacillary dysentery (BD) has brought a significant public health concern in China. Temperature is one of the main factors affecting BD incidence. Due to the largely different temperature ranges between regions, the classic multi-region time series studies could only explore the relative temperature-BD association and showed that BD incidence is positively associated with relative temperature (i.e., temperature percentile), which does not conform to the laboratory knowledge that both high and low temperature interfere with the survival of bacteria. The association on relative temperature scale also limits the intuition of epidemiological meanings.

### Methods

A novel two-stage strategy was proposed to investigate the association between monthly temperature and BD incidence on the original temperature scale in 31 provinces, China. In the first stage, truncated polynomial splines, as the substitute of the common natural splines or B-splines in generalized additive models, were used to characterize the temperature-BD association on the original temperature scale in each province. In the second stage, a multivariate meta-analysis compatible with missing values was used to pool the associations. The classic strategy based on relative temperature was used as a reference.

### Results

The average temperature-BD association presented a U-inverted shape, but not a monotonically increasing shape obtained using the classic strategy. This inverted U-shaped association conforms more to the laboratory knowledge and the original-scale association also provided an intuitive perspective and an easily explanatory result. Another advantage is that

replicated using the R codes via https://github.com/winkey1230/TS-based-strategy.

**Funding:** This work was supported by the non-profit Central Research Institute Fund of the Chinese Academy of Medical Sciences under Grant 2017PT32013. BZ received this funding. The funders had no role in study design, data collection and analysis, decision to publish, or preparation of the manuscript.

**Competing interests:** The authors have declared that no competing interests exist.

the novel strategy can extrapolate the province-specific association outside the observed temperature ranges by utilizing information from other provinces, which is meaningful considering the frequent incidences of extreme temperatures.

## Conclusions

The association between temperature and BD incidence presented a U-inverted shape. The proposed strategy can efficiently characterize the association between exposure and outcome on original scale in a multi-region study with largely different exposure ranges.

### Author summary

Temperature is one of the main factors affecting BD incidence. Due to the largely different temperature ranges between regions, the classic multi-region time series studies could only explore the relative temperature-BD association and showed that BD incidence is positively associated with relative temperature (i.e., temperature percentile), which does not conform to the laboratory knowledge that both high and low temperatures interfere with the survival of bacteria. The relative-scale association also limits the intuition of epidemiological meanings. In this study, we proposed a novel two-stage strategy to characterize the association between BD and original temperature in 31 provinces, China. We examined an inverted U-shaped association, but not a monotonically increasing shape obtained using the classic strategy. This new finding conforms more to the laboratory knowledge and the original-scale association also provided an intuitive perspective and an easily explanatory result. Another advantage is that the novel strategy can extrapolate the province-specific association outside the observed temperature ranges by utilizing information from other provinces, which is meaningful considering the frequent incidences of extreme temperatures.

## 1. Introduction

Bacillary dysentery (BD), an intestinal infectious disease caused by Shigella bacteria, is transmitted by the fecal-oral route through polluted food and water [1]. With approximately 188 million cases and 164,300 Shigella-related deaths reported per year worldwide [2], BD has become a considerable public health concern globally, especially in resource-poor countries [3]. In China, BD is the third-most common notifiable infectious disease.

Temperature is substantially associated with the transmission of enteric infections by affecting the survival of viruses and human behavior [4]. Several single-region studies have investigated the exposure-response relationship (ERR) between BD and temperature in different regions in China [5–9], but the results are inconsistent and not comparable due to the different analysis methods between regions. To obtain stronger evidences and comparable results in terms of BD-temperature ERRs, Liu et al. [10] conducted a nationwide multi-region study in China using the classic two-stage strategy by Gasparini et al. [11,12], which shows that temperature is positively associated with the risk of BD without any threshold in China. However, experiment studies show that both high and low temperatures can interfere the survival of Shigella bacteria [13], which does not conform to this observed epidemiological evidence. In addition, the obtained ERRs are measured on a relative temperature scale rather than the original scale, limiting the intuition of epidemiological meanings. These limitations are derived from

the usage of the classic two-stage strategy under the condition that temperatures present largely different ranges between regions.

Specifically, in the first stage, a unified generalized additive model (GAM) is constructed independently for each region to obtain rough region-specific ERRs. In the second stage, multivariate meta-analysis is used to obtain the average ERR and more stable region-specific ERRs by borrowing information from each other. This two-stage strategy has been widely used to explore the ERRs between environmental factors and health outcomes in multi-region studies [14–17]. To make use of the multivariate meta-analysis in the second stage, the forms of the commonly used natural splines or B-splines functions [18] defining ERRs in GAM must be identical among all regions, i.e., the knots and bounds must be set identical for generating basis variables with the same mathematical meanings among regions. However, different regions present largely different temperature ranges in China; if the identical knots and bounds are set based on abstract temperature, the generated basis variables are complete multicollinearity, making the parameters unidentifiable. Thus, temperature is scaled as region-specific percentile to unify the ranges across regions. As such, multivariate meta-analysis holds an assumption that the ERRs on the relative exposure scale present a similarity. Due to the human adaptability to environmental factors, such an assumption may be reasonable in some cases, for example, the impact of temperature on human mortality [19], but in many other cases, for example, the impact of temperature on the pathogenicity of virus and bacteria, this assumption may not be appropriate. The method using ERR on a relative scale seems to be more of a reluctant choice for most researchers.

As the observed ranges of exposure variables in most of multi-region studies across a large spatial scale are significantly different between regions and even without overlap, it is of important application values to develop a novel strategy to characterize the nonlinear ERRs on the original exposure scale in multi-region studies with largely different exposure ranges between regions. In this work, aiming to provide a strong and comparable nationwide evidence in China regarding the temperature-BD ERRs on the original temperature scale, we developed a novel two-stage strategy. In the first stage, the GAMs were constructed based on the truncated polynomial splines in which each knot independently defined a basis variable to avoid complete multicollinearity, thereby not confusing the parameter identification in the first stage. In the second stage, the multivariate meta-analysis compatible with missing values was used to pool the ERRs on the original scale for obtaining a more stable result. Another advantage is that our strategy provides an extrapolation of region-specific ERRs outside the observed exposure ranges by utilizing information from other regions, which is meaningful considering the frequent incidences of extreme temperature.

In Section 2 of the paper, we briefly introduced the classic two stage strategy and further justified its inability to investigate the ERR on the original exposure scale in a multi-region study with largely different exposure ranges. Based on the classic strategy, the novel two-stage strategy was then detailed. In Section 3, using the novel strategy, we conducted a nationwide study in 31 provinces of China to characterize the average and the province-specific ERRs on the original scale, which provided a novel perspective about the temperature-BD ERRs. The general discussion and conclusion are detailed in the last Section.

## 2. Methods

### Ethics approval and consent to participate

This is an observational study and the case data are freely available to the public. The Research Ethics Committee of Sichuan People's Hospital has confirmed that no ethical approval is required.

## 2.1 The classic two-stage strategy

In the first stage, for each region, a GAM is constructed using a general form:

$$g(\mu_{it}) = \alpha + s(x_{it}, \boldsymbol{\theta}_i) + confounders, \tag{1}$$

where $g(.)$ is a link function; $\mu_{it} \equiv E(y_{it})$ with $y_{it}$ being the observed numbers of outcomes in region $i$ at time $t$; $s(\cdot)$ is a nonlinear function; $x_{it}$ is the observed exposure variable; $\boldsymbol{\theta}_i$ is the unknown parameter vector defining the ERR in region $i$. The term, *confounders*, indicates the effects of confounding factors and the long-term trend. In the classic strategy, due to their smoothness and flexibility, natural splines and B-splines are often used to characterize the nonlinear ERRs, as such,

$$s(x_{it}, \boldsymbol{\theta}_i) = \sum_{j}^{df} \theta_{ij} b_j(x_{it}), \tag{2}$$

where $\theta_{ij}$ is the $j$th element of $\boldsymbol{\theta}_i$; $b_j(\cdot)$ is the $j$th basis function derived from natural splines or B-splines with prespecified attributions including knots, bounds, and degree of the polynomial; and $df$, degree of freedom, is the number of basis functions. Maximum likelihood method is used to obtain the unbiased estimation, $\hat{\boldsymbol{\theta}}_i$, along with its covariance $\hat{\mathbf{S}}_i$.

In the second stage, the pooled average ERR and more stable region-specific ERRs are obtained using a multivariate meta-analysis[11], which is constructed as follows:

$$\hat{\boldsymbol{\theta}}_i | \boldsymbol{\theta}_i \sim MN(\boldsymbol{\theta}_i, \hat{\mathbf{S}}_i), \boldsymbol{\theta}_i = \bar{\boldsymbol{\theta}} + \boldsymbol{\xi}_i, \boldsymbol{\xi}_i \sim MN(0, \boldsymbol{\psi}), \tag{3}$$

where $\boldsymbol{\theta}_i$ is the true ERR in region $i$; $\bar{\boldsymbol{\theta}}$ is the average ERR; $\{\boldsymbol{\xi}_i\}$ measures the heterogeneity of ERRs between regions; and $\boldsymbol{\psi}$ is the unknown between-region covariance matrix. Using maximum likelihood or restricted maximum likelihood to obtain the estimations of $\bar{\boldsymbol{\theta}}$ and $\boldsymbol{\psi}$, the best linear unbiased estimation of ERR in region $i$ is obtained as:

$$\hat{\boldsymbol{\theta}}_{b(i)} = \hat{\bar{\boldsymbol{\theta}}} + \hat{\boldsymbol{\psi}}(\hat{\mathbf{S}}_i + \hat{\boldsymbol{\psi}})^{-1}(\hat{\boldsymbol{\theta}}_i - \hat{\bar{\boldsymbol{\theta}}})$$

$$\operatorname{cov}(\hat{\boldsymbol{\theta}}_{b(i)}) = \operatorname{cov}(\hat{\bar{\boldsymbol{\theta}}}) + \hat{\boldsymbol{\psi}} - \hat{\boldsymbol{\psi}}(\hat{\mathbf{S}}_i + \hat{\boldsymbol{\psi}})^{-1}\hat{\boldsymbol{\psi}}.$$

When one needs to investigate the modification effects of region-level factors on ERR, the interesting factors can be incorporated into the multivariate meta-analysis, then the $\boldsymbol{\theta}_i$ in Formula (3) can be redefined as

$$\boldsymbol{\theta}_i = \mathbf{X}_i \boldsymbol{\beta} + \boldsymbol{\xi}_i, \tag{4}$$

where $\mathbf{X}_i$ is a matrix composed of the vector $\boldsymbol{x}_i$ referring to the region-level factors in region $i$, and $\boldsymbol{\beta}$ measures the strength of modification effect. When no region-level factor is included, $\boldsymbol{x}_i$ only indicates the intercept with a value of 1.

$$\mathbf{X}_i = \mathbf{I}_k \otimes \boldsymbol{x}_i^T = \begin{bmatrix} \boldsymbol{x}_i^T & \mathbf{0} & \cdots & \mathbf{0} \\ \mathbf{0} & \boldsymbol{x}_i^T & & \mathbf{0} \\ \vdots & & \ddots & \vdots \\ \mathbf{0} & \mathbf{0} & \cdots & \boldsymbol{x}_i^T \end{bmatrix},$$

$\mathbf{I}_k$ is a $k$-dimension identity matrix and $k$ is the dimension of $\boldsymbol{\theta}_i$. This methodology has been well demonstrated in Gasparrini et al.'s work [11].

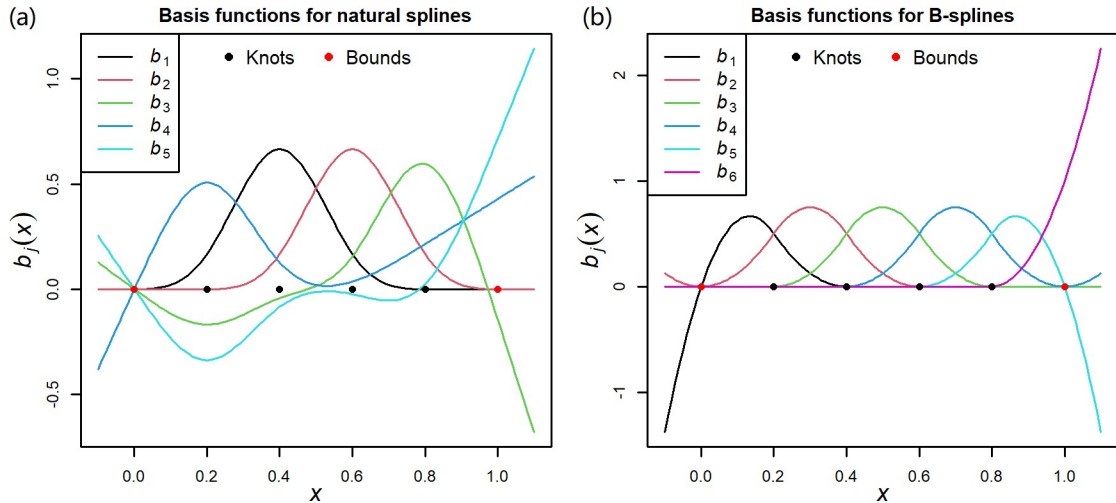

**Fig 1. The curves of basis functions in natural splines and B-splines.** Given four knots (0.2, 0.4, 0.6, and 0.8) and two bounds (0 and 1), $b_1-b_5$ in Panel a indicate the five basis functions whose linear combination forms the natural spline. The curves $b_1-b_6$ in Panel b indicate the six basis functions whose linear combination forms the B-spline function.

With further insight into the natural splines and B-splines, without loss of generality, the basis functions are illustrated in Fig 1 where the bounds (0 and 1) and four knots (0.2, 0.4, 0.6, and 0.8) are selected. The basis functions are jointly defined by multiple knots or bounds [20], which leads to the matrix composed of non-zero basis variables and an intercept of 1 being singular within a local range and further affects the parameter identification of $\boldsymbol{\theta}_i$. For example, within the local range of [0.2,0.6], the matrix composed of $b_1$, $b_2$, $b_3$, $b_4$, and 1 is singular for B-splines and the matrix composed of $b_1-b_5$ and 1 is also singular for natural splines. Therefore, if the exposure range in a study region is within [0.2,0.6], $\boldsymbol{\theta}_i$ defining the ERR cannot be estimated in the first stage, making the classic two-stage unavailable. This reason is detailed using R codes, which can be found at https://github.com/winkey1230/TS-based-strategy. Notably, although the singular matrix, i.e., complete collinearity, makes the coefficients of basis variables unidentifiable, in the R software the generalized linear model (GLM) excludes the colinear terms and sets the corresponding coefficients as missing for running codes available by default. If multivariate meta-analysis is forcibly used in the second stage without rationality, the information borrowing can be interpreted as mixing variables with different epidemiological meanings, which leads to biases.

## 2.2 A novel two-stage strategy

**2.2.1 The first stage.** Derived from the reason for generating the singular matrix as mentioned above, we used the truncated polynomial splines (TS) [21] to substitute the natural splines and B-spines in the first stage. Let

$$(z)_+ = \begin{cases} z, z > 0 \\ 0, z \leq 0 \end{cases} \text{ and } (z)_- = \begin{cases} z, z < 0 \\ 0, z \geq 0 \end{cases},$$

using the classic TS with the degree of polynomial equal to 2 and without intercept, Formula

(2) is expressed as:

$$s(x_{it}, \boldsymbol{\theta}_i) = \theta_1 x_{it} + \theta_2 x_{it}^2 + \sum_{j=1}^{K} \theta_{j+2} (x_{it} - \kappa_j)_+^2, \tag{5}$$

where $\kappa_1 > \kappa_2 > \cdots > \kappa_K$ are knots, and $x_{it}$, $x_{it}^2$ and $(x_{it} - \kappa_j)_+^2$ are the basis variables. As seen, apart from $x_{it}$ and $x_{it}^2$, each basis variable is defined by a single knot, thereby avoiding the singularity of the matrix in natural splines and B-splines. Specifically, when the exposure value in region $i$ is within a local range, such as $[\kappa_2, \kappa_{K-1}]$, the two basis variables, $(x_{it} - \kappa_{K-1})_+^2$ and $(x_{it} - \kappa_K)_+^2$, will be constantly zero, and then the TS in this region is determined as

$$s(x_{it}, \boldsymbol{\theta}_i) = \theta_1 x_{it} + \theta_2 x_{it}^2 + \sum_{j=1}^{K-2} \theta_{j+2} (x_{it} - \kappa_j)_+^2, \tag{6}$$

Although $\theta_{K+2}$ and $\theta_{K+1}$ cannot be identified, it does not affect the identification of other ERR-defining parameters in the local range of $[\kappa_2, \kappa_{K-1}]$. As such, the parameters defining such local ERR can be estimated in GAM as

$$\hat{\boldsymbol{\theta}}_i = (\hat{\theta}_1, \hat{\theta}_2, \cdots, \hat{\theta}_K, NA, NA)^T, \tag{7}$$

$$\text{cov}(\hat{\boldsymbol{\theta}}_i) = \begin{bmatrix} a_{11} & \cdots & a_{1K} & NA & NA \\ \vdots & \ddots & \vdots & \vdots & \vdots \\ a_{K1} & \cdots & a_{KK} & NA & NA \\ NA & \cdots & NA & NA & NA \\ NA & \cdots & NA & NA & NA \end{bmatrix}, \tag{8}$$

where *NA* indicates the value is missing and not available. Unlike the missing coefficients for B-splines or natural splines mentioned in the last part of Section 2.1, the missing coefficients for TS are set for the reason that the corresponding basis variables are constantly zero and provide no identifiable information, so they do not confuse the other coefficients and the problem of mixing variables with different epidemiological meanings in the meta-analysis can be efficiently addressed.

Furthermore, since the exposure data are more easily concentrated on the center in multi-region studies, the following extended TS may be a more appropriate choice:

$$s(x_{it}, \boldsymbol{\theta}_i) = \theta_1 x_{it} + \theta_2 x_{it}^2 + \sum_{j=1}^{m} \theta_{j+2} (x_{it} - \kappa_j)_-^2 + \sum_{j=m+1}^{K} \theta_{j+2} (x_{it} - \kappa_j)_+^2, \tag{9}$$

where $\kappa_m$ and $\kappa_{m+1}$ are the lower centered and upper centered knots, respectively. Considering 0.2, 0.4, 0.6, and 0.8 as the knots, with 0.4 and 0.6 as the centered knots, the basis functions of the TS are intuitively presented in Fig 2. 

When the degree of the polynomial is equal to 2, the TS is differentiable of the first order everywhere and is smooth enough to characterize the ERRs in environmental health. When the degree of the polynomial is equal to 3, the TS is differentiable of the second order everywhere, and can be written as:

$$s(x_{it}, \boldsymbol{\theta}_i) = \theta_1 x_{it} + \theta_2 x_{it}^2 + \theta_3 x_{it}^3 + \sum_{j=1}^{m} \theta_{j+3} (x_{it} - \kappa_j)_-^3 + \sum_{j=m+1}^{K} \theta_{j+3} (x_{it} - \kappa_j)_+^3$$

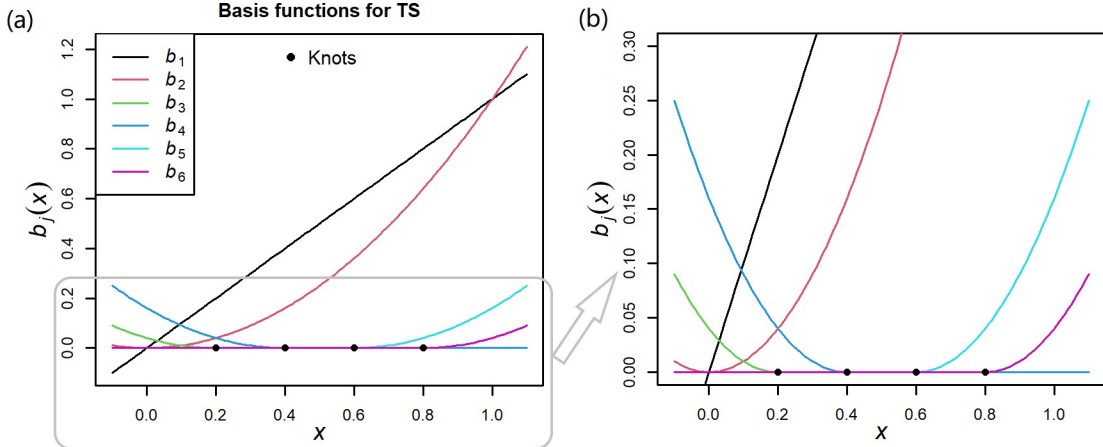

**Fig 2. The curves of basis functions in the TS.** The knots were set as 0.2, 0.4, 0.6, and 0.8, where 0.4 and 0.6 were the centered knots. The degree of the polynomial was 2. Panel (b) is the magnification of Panel (a) for a local area.

**2.2.2 The second stage.** Given the fixed knots on the original exposure scale, based on the TS, the rough region-specific ERRs, $\{\hat{\boldsymbol{\theta}}_i\}$ possibly with missing elements as in Eqs (7) and (8), are obtained in the first stage. To make the multivariate meta-analysis available in the second stage, the missing elements in $\hat{\boldsymbol{\theta}}_i$ are initially set as zero with large uncertainty, i.e., a non-information prior is assigned to the missing elements. For instance, the values in Eqs (7) and (8) can be set as

$$\hat{\boldsymbol{\theta}}_i = (\hat{\theta}_1, \hat{\theta}_2, \cdots, \hat{\theta}_K, 0, 0)^T, \tag{10}$$

$$\mathrm{cov}(\hat{\boldsymbol{\theta}}_i) = \begin{bmatrix} a_{11} & \cdots & a_{1K} & 0 & 0 \\ \vdots & \ddots & \vdots & \vdots & \vdots \\ a_{K1} & \cdots & a_{KK} & 0 & 0 \\ 0 & \cdots & 0 & 10^{10} & 0 \\ 0 & \cdots & 0 & 0 & 10^{10} \end{bmatrix}. \tag{11}$$

Subsequently, the classic multivariate meta-analysis can be used as usual. In practice, the multivariate meta-analysis also directly supports the vectors with missing elements and has been implemented in the frequently used R package 'mvmeta' [11].

Using the multivariate meta-analysis, not only can we obtain the parameter $\bar{\boldsymbol{\theta}}$ defining the average ERR and improve the stability of estimation of region-specific ERRs on the original exposure scale, but can also estimate the missing elements in $\hat{\boldsymbol{\theta}}_i$ by borrowing information from other regions where the corresponding estimated parameters are non-missing. So, for region $i$, the ERR within a local range can be further extrapolated to the entire range in this study, which is meaningful when a historically unparalleled exposure event occurs in a specific region.

## 2.3 Data

A total of 31 provinces in mainland China were selected. The spatial distribution of provinces is shown in Fig 3A. For each province, the monthly numbers of BD cases from 2004 to 2017 were retrieved from the Chinese Center for Disease Control and Prevention via http://en.

chinacdc.cn/. A total of 3,710,962 BD cases, with an average of 712,55 cases per month, were obtained. The province-level daily temperature data were collected from the China Meteorological Data Sharing Service System (http://cdc.cma.gov.cn/), based on which the population-weighted monthly province-level temperatures were calculated as the exposure variable. In addition, because previous studies showed that relative humidity and sunshine duration had a significant impaction on BD incidence [6,22], we also calculated the population-weighted monthly province-level relative humidity and sunshine duration as confounders. The spatial distributions of average temperatures and total cases of BD in mainland China from 2004 to 2017 are presented in Fig 3B and 3C, respectively. As shown in Fig 3D, the BD cases substantially decreased over time and the short-term fluctuation was aligned with that of temperatures. Fig 3E presents the ranges of monthly temperature in 31 provinces. As seen, these ranges were largely different among provinces and even hardly overlap in Xizang and Hainan, which suggested that the classic two-stage strategy could only obtain the ERRs on the relative scale.

## 3. Results

### 3.1. Characterizing the BD-temperature ERRs on the original scale

We used the proposed novel two-stage strategy to investigate the ERRs on the original scale. In the first stage, the TS-based GAM was constructed as:

$$y_{it} \sim Quasi - Poisson(\mu_{it})$$

$$\ln(\mu_{it}) = \alpha_i + TS(tem_{it}, \boldsymbol{\theta}_i) + s(humid_{it}) + s(sunshine_{it}) + s(time_{it}) \tag{12}$$

where $y_{it}$ is the number of BD cases in province $i$ at time $t$. Quasi-Poisson distribution was used to characterize the over-dispersion. The parameters were selected based on the Akaike information criterion (AIC). Specifically, for TS, the degree of the polynomial was set as 2 and the knots were set as 10%, 30%, 60%, and 90% quantile temperatures across all the province-level monthly temperatures as in Fig 3E, corresponding to -2, 8.4, 18.5, and 26.2°C, respectively, in which 8.4 and 18.5 (°C) are the centered knots. The confounding effects of humidity and sunshine duration were adjusted for using linear associations. The long-term trend was characterized by using natural cubic splines with a degree of freedom of 6. In the second stage, the multivariate meta-analysis with missing values was used to pool the $\{\hat{\boldsymbol{\theta}}_i\}$ estimated in the first stage.

Taking the median temperature of 15.3°C across all the province-level monthly temperatures as a reference, on the original exposure scale, the curves of the pooled average BD-temperature ERRs are presented in Fig 4A. Overall, when the monthly temperature was below 28°C, a high temperature was associated with a high BD risk. When monthly temperature was above 28°C, the risk started to decrease. Within the range of 10–25°C, the BD risk was sensitive to temperature variation. Different provinces presented substantially heterogeneous BD-temperature ERRs, especially within a high-temperature range. Besides, we also presented the province-specific ERRs within the overall temperature range to provide more detailed information, as in Fig 5. Taking the ERR in Beijing as an example, a high temperature was associated with a high BD risk within the observed temperature range. When the temperature was below the minimal observed temperature, the BD risk continued to decline, but when it was above the maximal observed temperature, the BD risk no longer increased. In a multi-region study with much different exposure ranges, it was not feasible for the classic two-stage strategy to extrapolate the ERR outside the observed exposure range, while such extrapolation was meaningful in the context where there was an increased frequency of emergence of extreme

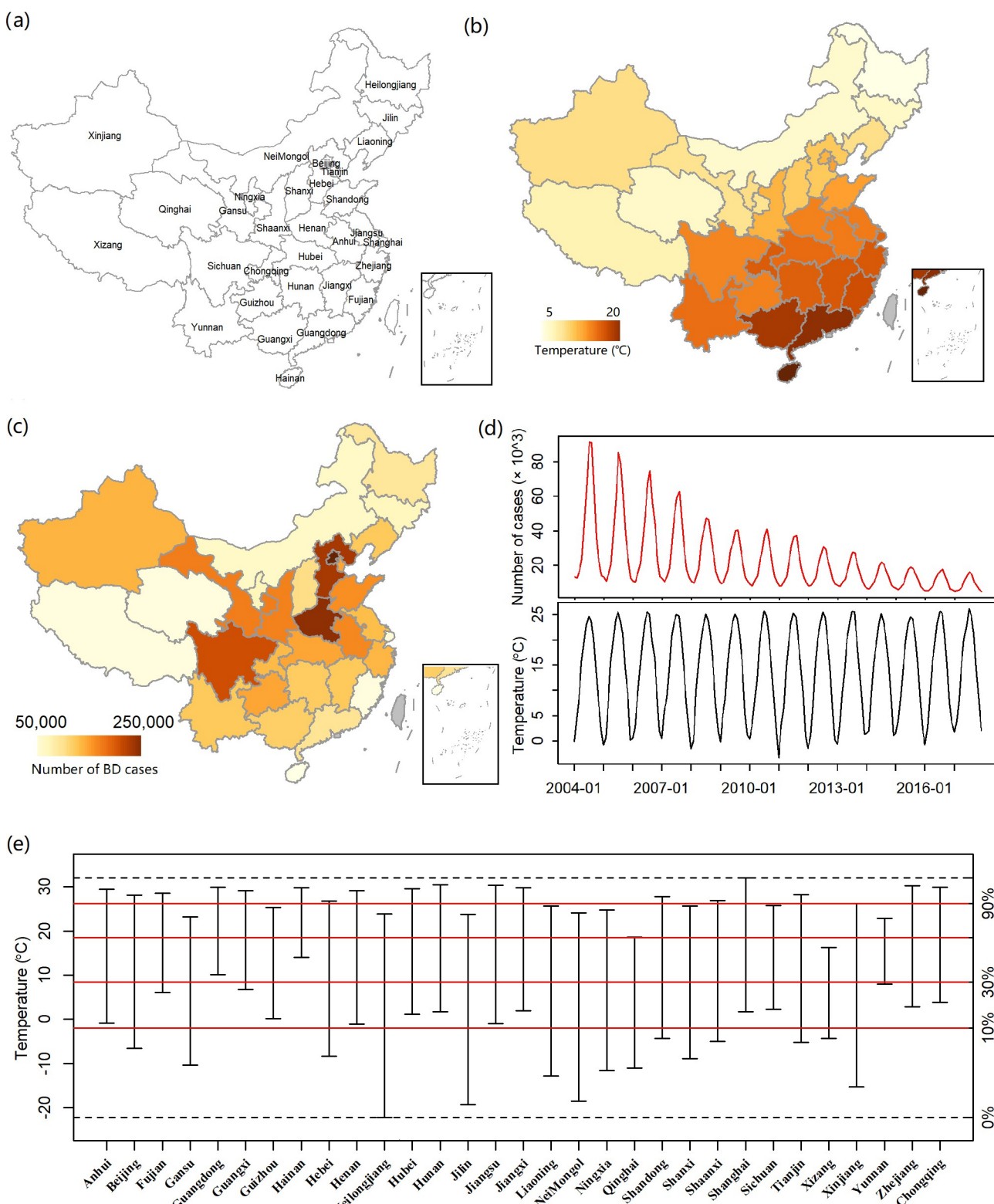

**Fig 3. Descriptive analysis for temperature and BD cases.** Panel (a) presents the locations of 31 provinces in China. Panels (b) and (c) present the spatial distributions of average temperatures and total cases of BD in mainland China from 2004 to 2017, respectively. Panel (d) present the temporal trends of the total BD cases and average temperatures. Panel (e) presents the temperature ranges in 31 provinces, where the red horizonal lines indicate the 10%, 30%, 60%, and 90% quantile temperatures across all the province-level monthly temperatures. The maps were created using the "tmap" (v3.3–2) package for R software (v4.1.1). The used base map was from http://bzdt.ch.mnr.gov.cn/.

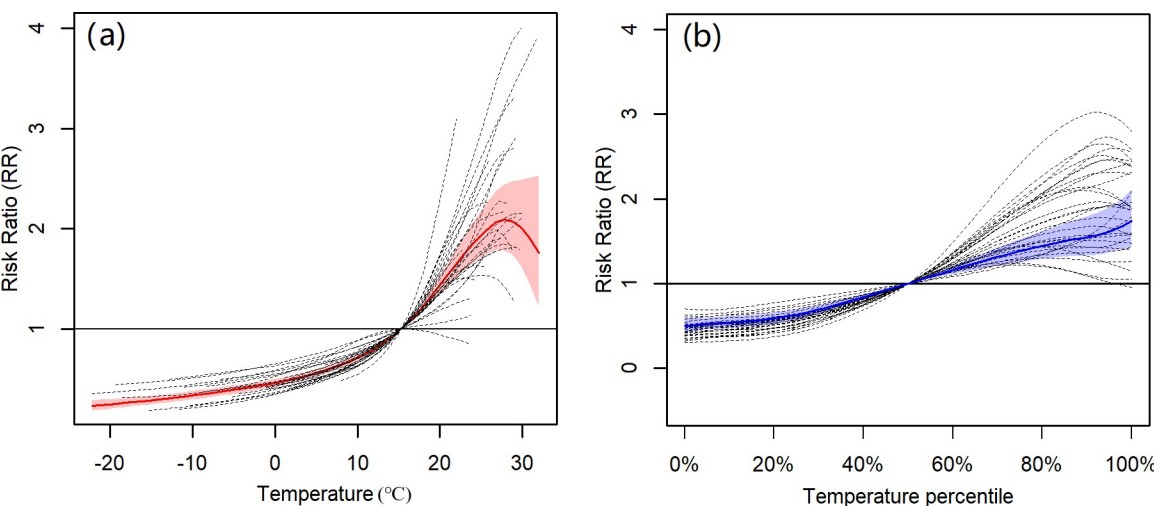

**Fig 4. The average BD-temperature ERRs.** Panel (a) presents ERRs on the original scale from the TS-based strategy. Panel (b) presents ERRs on the relative scale from the B-spline-based strategy. The solid line is the average ERR and the shaded area is 95% confidence interval. The dashed lines are the province-specific ERRs. ERR: exposure-response relationship.

environmental events, such as temperature and air pollutants [23–25]. In several provinces (e.g., Yunnan and Xizang), the extrapolated RRs at high temperatures were substantially higher than those in other provinces and even exhibited no decrease tendency, but which may not be believable due to the wide confidence intervals. In contrast, the extrapolated associations at low temperature are believable because of the narrow confidence intervals.

For sensitivity analyses, on the basis of main analysis, we changed the knots to 10%, 50%, and 90% quantile temperatures with the latter two as the centered knots (Sensitivity analysis 1), changed the *df* of long-term trend to 4 (Sensitivity analysis 2) and 8 (Sensitivity analysis 3), and changed the centered knots to 10% and 30% quantile temperatures (Sensitivity analysis 4). In addition, the sunshine duration may be deemed as part of the effects of temperature, thus we also excluded this variable in model 11 (Sensitivity analysis 5). The results in sensitivity analysis remained stable, which are detailed in the supplementary materials (Figs A-N in S1 Text). Although the B-splines-based two-stage strategy mentioned in the last part of Section 2.1 suffers from the mixing of different information and has never been applied in practice, we still used it to characterize the ERRs on the original scale as a comparison. As expected, the B-spline-based method gave odd temperature-BD associations seen in Fig O in S1 Text, which did not conform to the existing epidemiological evidence that low temperature will decrease the BD risk.

### 3.2. Exploring the modification effects and spatial heterogeneity

Based on the estimated province-specific associations in the first stage, we incorporated a series of province-level predictors, such as relative humidity, poverty, education, GDP and so on, into the multivariate meta-regression to test the potential effect modifiers which may impact the temperature-BD associations. The poverty is measured by the proportion of people whose family incomes are below the lowest living standard and the education is measured by the number of pupils per teacher. More detailed information of predictors can be seen in Section 1.1 of the supplementary materials. Due to their significant spatial dependences, the incorporated predictors can control for the spatial confounding in the multivariate meta-regression to some extent. In addition, we also used the multivariate conditional meta autoregression [26]

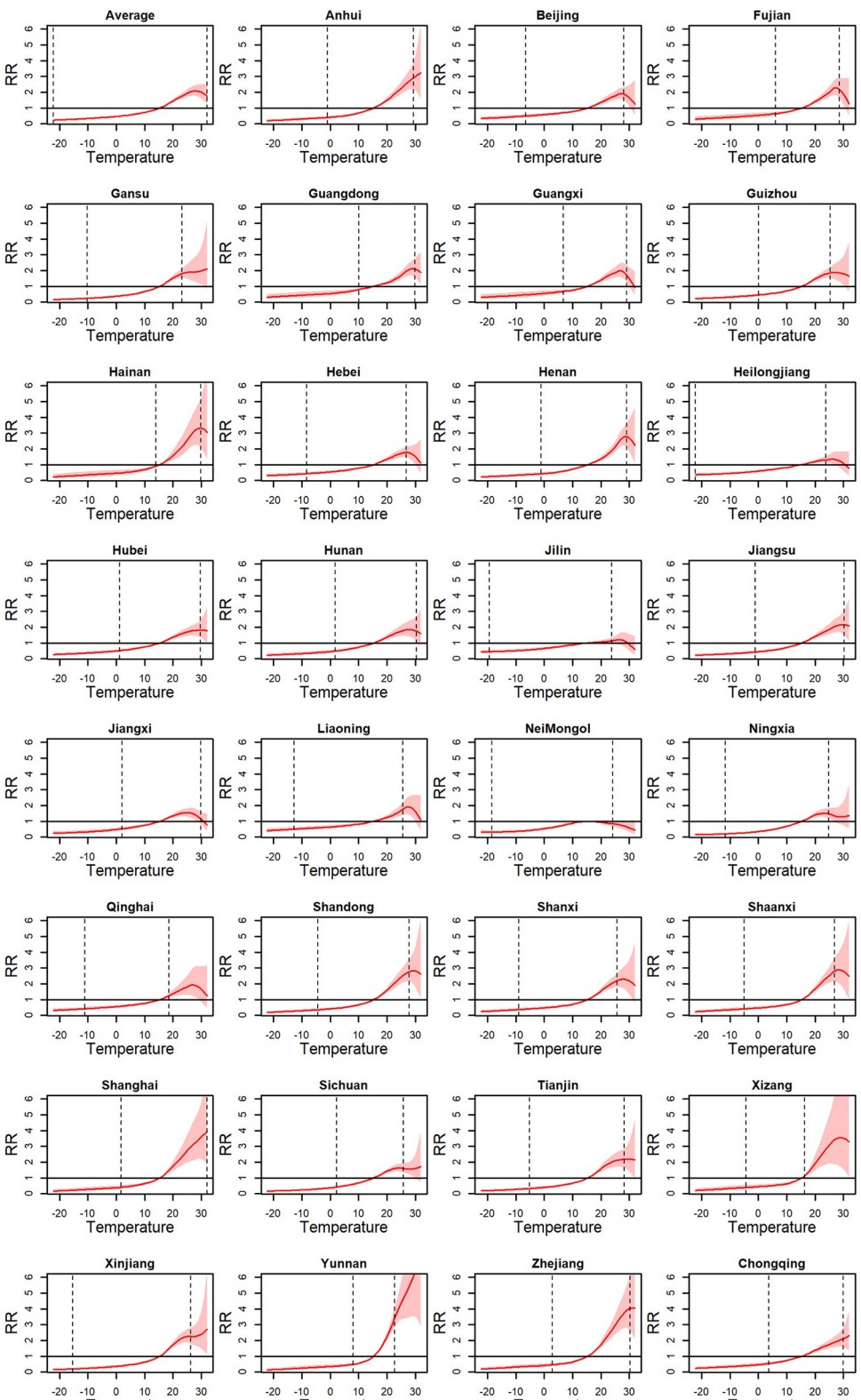

**Fig 5. The province-specific BD-temperature ERRs on the original scale using the TS-based strategy.** The vertical dashed lines indicate the observed province-specific bounds of temperature. The overall median temperature of 15.3°C was the reference. ERR: exposure-response relationship.

**Table 1. The identification of effect modifiers and heterogeneity tests in multivariate meta-regressions.**

| Predictors [a] | AIC | LR [b] | | | Q test [c] | | | I² |
|---|---|---|---|---|---|---|---|---|
| | | Stat | df | P | Stat | df | P | |
| Intercept | -239.6 | - | - | - | 875.1 | 180 | < 0.001 | 79.4 |
| Rainfall | -241.8 | 14.2 | 6 | 0.028 | 802.7 | 174 | < 0.001 | 78.3 |
| Relative humidity | -238.7 | 11.1 | 6 | 0.086 | 820.6 | 174 | < 0.001 | 78.8 |
| Sunshine duration | -237.4 | 9.8 | 6 | 0.135 | 813.1 | 174 | < 0.001 | 78.6 |
| Per-capital GDP | -236.2 | 8.5 | 6 | 0.203 | 851.7 | 174 | < 0.001 | 79.6 |
| Per-capital wage | -234.1 | 6.4 | 6 | 0.375 | 860.1 | 174 | < 0.001 | 79.8 |
| Urbanization rate | -235.1 | 7.4 | 6 | 0.284 | 850.2 | 174 | < 0.001 | 79.5 |
| Poverty | -247.9 | 20.2 | 6 | 0.003 | 769.7 | 174 | < 0.001 | 77.4 |
| Education | -239.7 | 12 | 6 | 0.062 | 768.4 | 174 | < 0.001 | 77.4 |
| Doctors | -234.1 | 6.4 | 6 | 0.376 | 837.6 | 174 | < 0.001 | 79.2 |
| Health-related worker | -233.4 | 5.7 | 6 | 0.452 | 851.9 | 174 | < 0.001 | 79.6 |
| Hospital beds | -235.7 | 8 | 6 | 0.236 | 834.3 | 174 | < 0.001 | 79.1 |
| Poverty + education | -250.4 | 34.8 | 12 | < 0.001 | 693.1 | 168 | < 0.001 | 75.8 |

Note

[a] The detailed explanations for the predictors can be seen in Text A in S1 Text. Intercept indicates that the multivariate meta-regression includes only an intercept and no predictor

[b] The likelihood ratio test for $\beta$ in Formula (4) was used to test if a predictor significantly modifies the association

[c] I² statistic was used to measure the residual heterogeneity and Cochran Q test was used to test the residual heterogeneity.

to test the residual spatial dependence in the meta-regression and results showed that there is no significant residual spatial dependence, suggesting that the spatial confounding was well controlled.

Shown in Table 1, the I² statistics showed that a considerable heterogeneity exists among provinces. The meta-regressions only with one predictor identified the poverty and rainfall as significant modifier effects ($P < 0.05$). The $P$ values of relative humidity and education were closed to 0.05. After using an AIC-based forward method to select variables, only poverty and education were included into the multiple meta-regression with P < 0.001, suggesting that poverty and education were two important factors of affecting the temperature-BD associations, which conforms to the expectation since BD is a poverty-related infectious disease [3]. Furtherly, based on the multiple meta-regression, we plotted the temperature-BD associations at high/low poverty and good/poor education. Shown in Fig 6, the four association curves presented a U-inverted shape; especially at low poverty and good education, the U-inverted shape was obvious. For the associations at high poverty and poor education, the risk of BD rose as temperature increased until temperature was above 29°C. In addition, by comparing the slopes in the association curves, the BD incidence at good education seems to be less sensitive to temperature than that at poor education. The reason may be that people with good education have more temperature-related consciousness of protection, thus reducing the impact of temperature on BD incidence.

According to the climate and geography, we also divided the study region into 7 zones and plotted the average association in each zone. Seen in Fig 7, because the subgroup analysis blocks the information borrowing between zones, we obtained the association curve only within the observed temperature range for a specific zone. Overall, the BD incidence rose as temperature increased in all the zones, but when temperature was above 28–30°C, apart from the southwest of China, the BD incidence no longer rose and even started to decline especially in the east of China. In the northeast, the association curve presented a flatter slope, suggesting

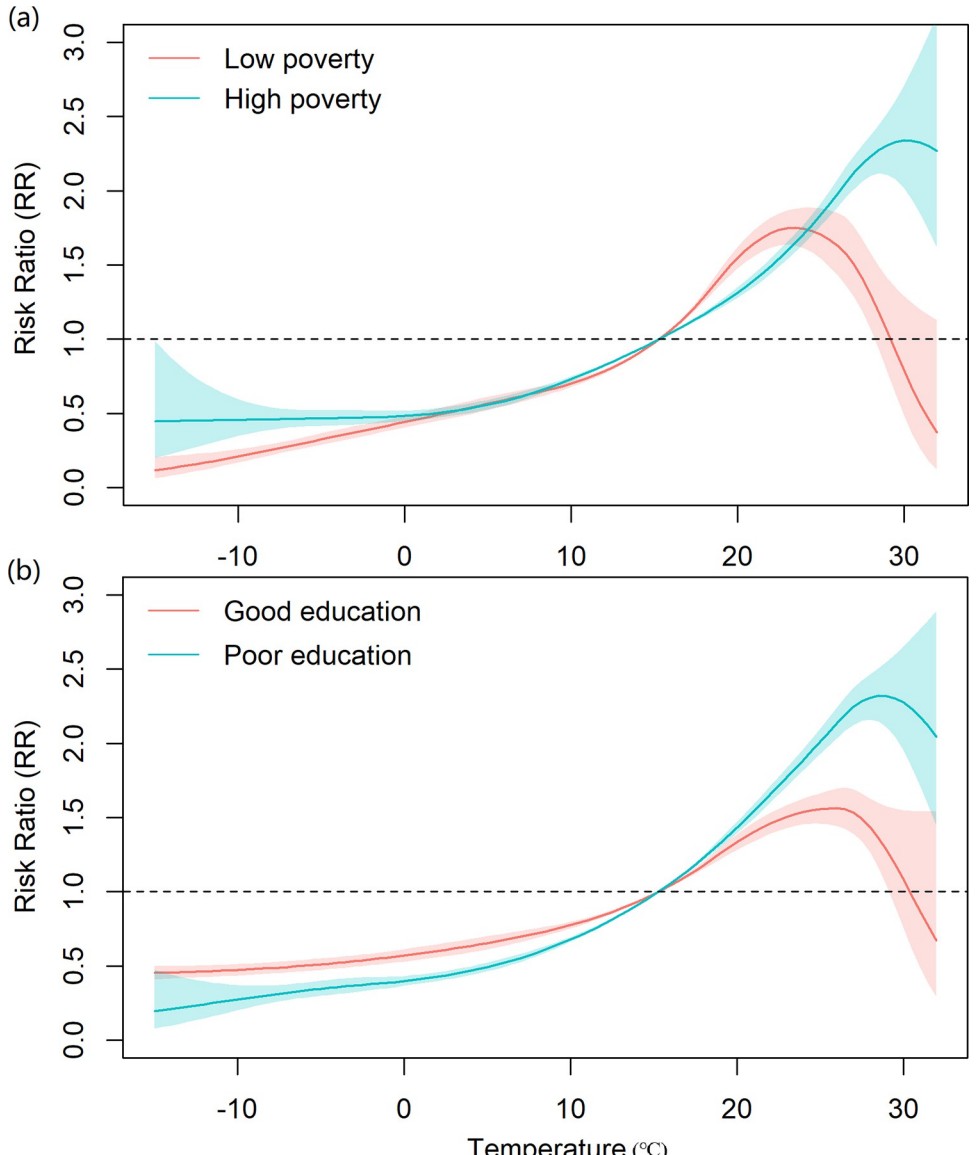

**Fig 6. The temperature-BD associations at different poverty and education.** The association curves come from the multivariate meta-regression with poverty and education as predictors. The high and low poverties refer to the proportions of people whose family incomes are below the lowest living standard as 0.135 and 0.007, respectively. The good and poor educations refer to the number of pupils per teacher as 11 and 20, respectively.

that the BD incidence is less sensitive to temperature than those in other zones; we did not obtain its association at a high temperature (>25°C) due to the observed temperature ranges less than 25°C.

## 3.3. Comparing ERRs on the relative scale with those on the original scale

For more clarity on the difference between the ERRs on the relative scale and those on the original scale, the classic two-stage strategy was also used to characterize the BD-temperature ERRs on the relative scale. In the first stage, a B-splines-based GAMs was constructed for each province. The degree of the polynomial was also set as 2; the bounds were set as the maximal

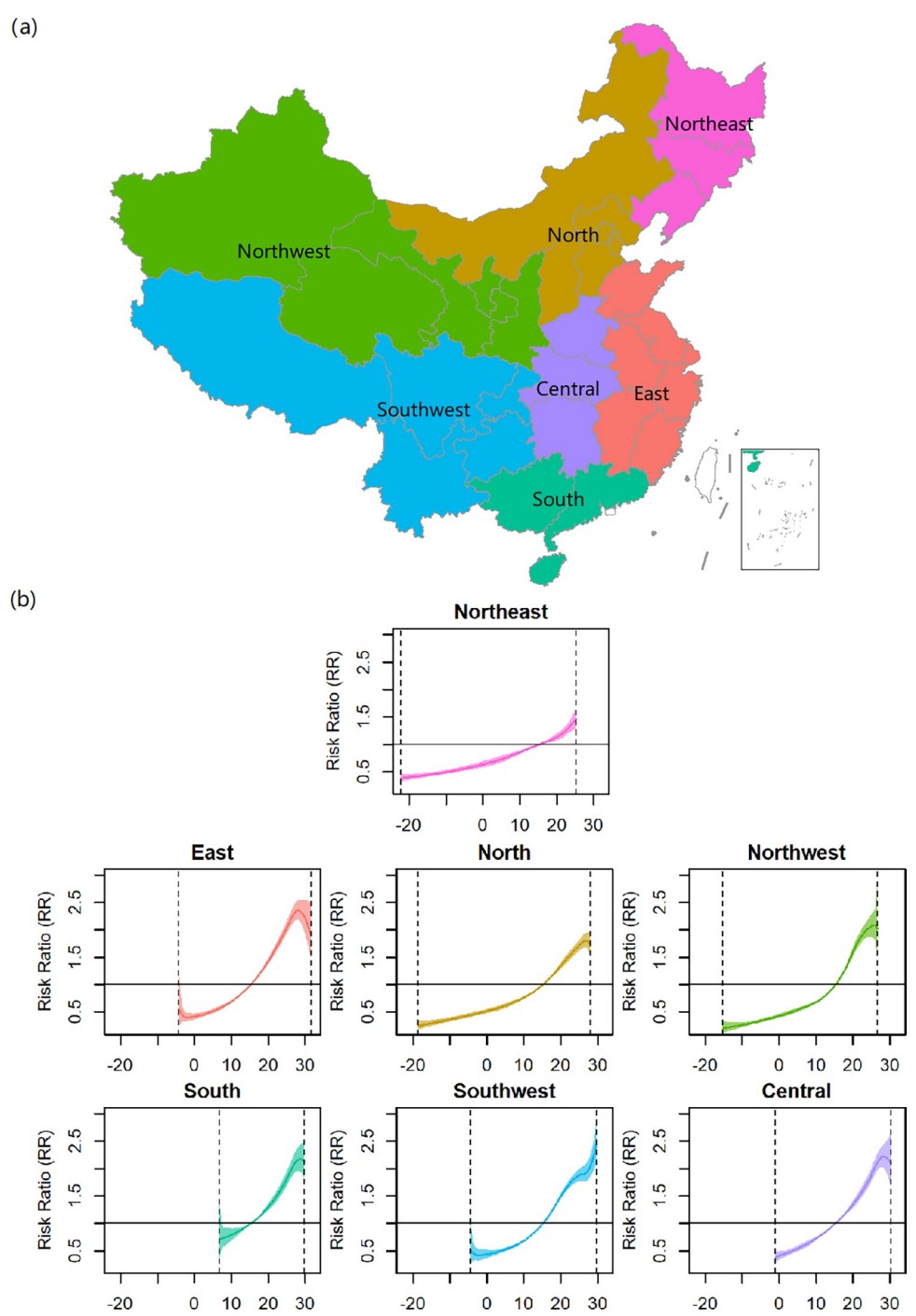

**Fig 7. The associations between temperature and BD incidence in the seven zones.** The maps were created using the "tmap" (v3.3–2) package for R software (v4.1.1). The used base map was from http://bzdt.ch.mnr.gov.cn/.

and minimal observed temperatures within a province and the within-province quantiles of temperature were set as 10%, 30%, 60%, and 90%. The confounders and the long-term trend were adjusted for using the same method with the TS-based strategy. In the second stage, the

common multivariate meta-analysis was used to pool the ERRs. As such, information borrowing in multivariate meta-analysis was employed for ERRs on the relative scale.

For province-specific ERRs, the within-province 50% quantile temperature was used as a reference and the temperature range was the observed maximal and minimal within-province temperatures. For the average ERR, the average within-province 50% quantile temperature across the 31 provinces was used as a reference and the bounds of temperature was the average within-province observed minimal and maximal temperatures. The comparison of the estimated ERRs using the classic and novel two-stage strategies is shown in Figs 8 and 4. Since any percentile corresponded to a single temperature in a specific province, the rescaling of the province-specific ERRs from the relative scale to the original scale was available in the classic strategy, seen in Fig 8. But for the average ERRs, the rescaling may not be intuitive since a percentile corresponded to different temperatures in different provinces. Therefore, the average ERR from the classic strategy could only be explained on the relative scale, limiting the intuition and the epidemiological meanings. Besides, although the temperature range of the ERRs from the novel strategy is larger than that from the classic strategy, the province-specific ERRs from the classic strategy are similar to those from the novel strategy within the observed temperature range. Despite a high degree of similarity in province-specific ERRs, the average ERR differs significantly between the two strategies, particularly in high temperatures. Specifically, the classic strategy examined a monotonically increasing shape for the average ERR on the relative scale, while the novel strategy examined a U-inverted shape for the average ERR on the original scale. The epidemiological rationality is discussed in Section 4.

Previous studies have shown that the TS has a good ability to fit nonlinear associations [21], but it is rarely used in environmental epidemiology. Therefore, we also used a multi-region study with slightly different exposure ranges to compare the TS-based strategy with the B-spline-based strategy in the context with an identical purpose, i.e., characterizing ERRs on the original scale. This example was based on the work by Gasparini et al. via https://github.com/gasparrini/2012_gasparrini_StatMed_Rcodedata[11], which has been used to illustrate the classic two-stage strategy. This dataset included the time series data of daily temperatures and death counts for 10 regions in England and Wales during 1993–2006. Results showed that both strategies could obtain region-specific ERRs outside the observed temperature ranges. The point estimations of ERRs and the 95% confidence intervals were almost the same between the two strategies, which suggested that the TS-based and the B-splines-based strategies have similar abilities in characterizing the nonlinear ERRs. The detailed results can be seen in Figs P-R in S1 Text of the supplementary materials.

## 4. Discussion

In a multi-region study where the ranges of exposure variable present largely difference between regions, the classic two-stage strategy can only be employed to explore nonlinear ERRs on a relative exposure scale, which may limit the intuition of epidemiological meanings, especially for the average ERR. In this work, we proposed a novel two-stage strategy based on truncated polynomial splines and multivariate meta-analysis with missing values to address the issue. This novel strategy cannot only explore ERRs on the original scale for providing a new perspective, but also can extrapolate region-specific ERRs outside the observed exposure ranges by borrowing the information from other regions.

In the example of investigating BD-temperature ERRs, the novel two-stage strategy performed well, as expected. Due to the largely different temperature ranges between provinces, this work is the first nationwide multi-region study to characterize the BD-temperature ERRs on the original scale. Results showed that the average ERR presented a U-inverted shape. In

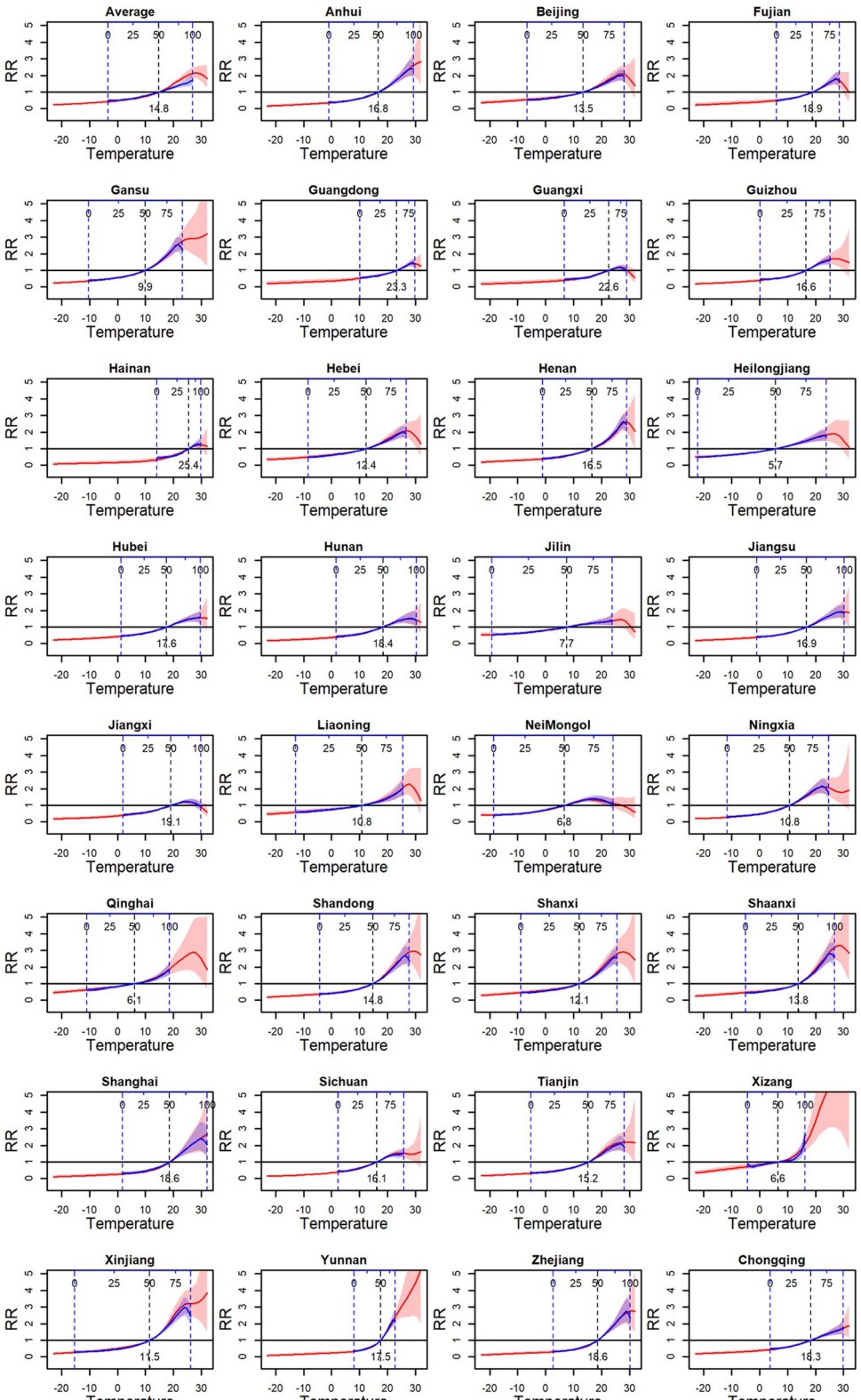

**Fig 8. The comparison of ERRs on the original scale with those on the relative scale.** The red solid curves indicate the ERRs on the original scale using the TS-based strategy. The blue solid curves indicate the ERRs on the relative scale using the B-spline-based strategy. The top axes indicate the relative scale. The vertical blue dashed lines indicate the

observed within-province minimal and maximal temperatures. The vertical black dashed lines indicate the reference temperature, i.e., within-province 50% quantile temperatures.

more details, when the monthly temperature was less than 28°C, a high temperature was associated with a high BD risk, but when it was above 28°C, the risk no longer continued to increase and even started to decrease. This U-inverted shape is more consistent with the laboratory knowledge [13] that high temperature can interfere with the survival of Shigella bacteria and many daytime temperatures in summer are over the optimal survival temperature. Our new finding provided a better understanding regarding the temperature-BD association and may help design a more rational temperature-related public health policy in BD prevention and control. On the other hand, in most of provinces with maximum monthly temperature above 28°C, the province-specific ERRs also present a U-inverted shape (Fig 4A), suggesting the rationality of U-inverted average ERR. In contrast, for the ERRs in the relative scale from the classic two-stage strategy, many provinces also presented a U-inverted shape (Fig 4B), but the average ERR is not. The reason is that the ERRs within high temperature ranges and the ERRs within low temperature range are synthesized in the multivariate meta-analysis; with a significantly monotone increasing ERRs within low temperature ranges, the synthesized average ERR tended to be more similar with the ERRs in low-temperature-range provinces than those in high-temperature-range provinces, thus presenting a monotone increasing shape. Such monotone association is consistent with those in previous nationwide studies [10,27] where the association was also characterized on the relative exposure scale. This result suggests that if the novel strategy is used to recharacterize the temperature-BD associations alternatively on the original temperature scale in these studies, a different association may be observed.

Another interesting thing is that the two strategies obtained similar province-specific ERR within the observed within-province temperature range, but the novel strategy can additionally extrapolate the ERRs outside the observed province-specific temperature ranges, which is meaningful considering the frequent occurrences of extreme temperature events in the background of global warming [24,28]. On the other hand, because the monthly temperature fluctuates slightly even in an extreme situation, the extrapolated association at substantially low temperatures in some provinces (e.g., Guangxi) may have a limited public health importance. Alternatively, if one investigated the associations using multi-region daily data, such as in Liu et al. [10], and Hao et al.'s works [29], the extrapolation of our method may deserve a more public health importance due to the larger fluctuation in daily temperatures. Notably, the extrapolation is resulted from borrowing information from other regions, not violating the attribute that spline functions lack extrapolation ability. Therefore, the exposure range for extrapolating ERRs is not supposed to be outside the overall range across all the observed exposure values, due to the absence of exposure data outside this overall range.

In the classic two-stage strategy, when a set of fixed knots are prespecified in a multi-region study with largely different exposure ranges, the frequently used B-splines and natural splines generate non-zero basis variables with complete multicollinearity, which affects the identification of parameters. In contrast, for the TS in the novel strategy, each knot defines a basis variable by itself to avoid complete multicollinearity. Therefore, even though some zero-constant basis variables are generated, the regression coefficients of non-zero basis variables can still be identified in the first-stage models. The identified local regression coefficients exactly define the ERR within a local exposure range. Then, in the second stage, a non-information prior is assigned to the missing coefficients, which makes all the properties of multivariate analysis available as in the classic strategy. As seen, the key to the success of the novel strategy is adapting a spline function to avoid complete multicollinearity among non-zero basis variables. So,

with the condition that each knot defines a basis variable by itself, some other spline functions, such as radial basis [21], can also be used to substitute the TS in the first stage model, at least from the statistical perspective.

In a multi-region study with largely different exposure ranges, the essential difference between the classic and novel two-stage strategies is that the former worked on the assumption of the between-region similarity in ERRs on the relative scale, but the latter worked on the assumption of the similarity in ERRs on the original scale. In most practical studies, it is difficult to decide the more appropriate assumption among the two. In the absence of a prior, we proposed to employ the novel two-stage strategy since it provides a more intuitive epidemiological explanation and has the ability to extrapolate the region-specific ERRs outside the observed exposure ranges without any loss of advantage. Another thing worth noting is that, as in Eq (4), the common TS does not set the centered knots, which may lead to an unsatisfactory result in some regions. For example, as described in the sensitivity analysis 4 of the supplementary materials, in Hainan province where the temperature range was higher than the 30% overall quantile temperature of 8.4˚C, the obtained 95% confidence interval of ERR was largely wide, thus resulting in the loss of statistical power. Alternatively, the proposed extended TS with centered knots can significantly improve the results. Besides, the values of AIC also showed that the TS with centered knots was a better choice than that without centered knots. Therefore, it was necessary to select the centered knots in a practical multi-region study. When fixing the knots, we proposed to select the two knots whose interval includes the exposure values as many regions as possible.

Previous studies have showed that the TS has a good ability to fit the nonlinear curve, but it is rarely used in environmental epidemiology. So, we further adopted a common multi-region study with slightly different exposure ranges to compare the TS-based novel strategy with the B-splines-based classic strategy. Results showed that the estimated ERRs were almost similar between the two strategies, suggesting that the TS and B-splines have a similar ability to fit nonlinear associations also in environmental epidemiology. In addition, we also compared the TS-based novel strategy with the natural-splines-based classic strategy, results showed that when the exposure was extremely high or low, the ERRs obtained by the two strategies were a little different, but the overall similarity was acceptable, which is explained in the supplementary materials.

Since the maximum incubation period of BD is shorter than one month [30], the lag effect of temperature on BD cannot be considered in the motivating example using monthly data. When the lag effect needs to be considered, a distributed lag nonlinear model (DLNM) [31,32] can be constructed in the first stage. Due to the similar statistical properties, the proposed strategy can be easily extended to DLNM-based two-stage strategy. Furthermore, if the ERRs present a strong spatial correlation or spatial structure, the multivariate meta-analysis with a spatial structure in the second stage [26,33–35] can be easily compatible with the novel strategy to achieve a better result.

## 5. Conclusions

The temperature-BD association presented a U-inverted shape, but not a monotonically increasing shape obtained using the classic strategy. This U-inverted shape conforms more to the laboratory knowledge and the association on the original scale also provided an intuitive perspective and an easily explanatory result. Another advantage is that the novel strategy can extrapolate the province-specific association outside the observed temperature ranges by utilizing information from other provinces, which is meaningful considering the frequent incidences of extreme temperatures. The proposed strategy can efficiently characterize the

association between exposure and outcome on the original scale in a multi-region time series study with largely different exposure ranges. The proposed technique would have a broad range of applications given the large number of such multi-region studies that have been and are being conducted.

## Supporting information

**S1 Text. Fig A. The curves of pooled average and province-specific BD-temperature ERRs in sensitivity analysis 1.** Fig B. The pooled average and province-specific ERRs in the overall temperature ranges in sensitivity analysis 1. Fig C. The comparison of ERRs between the classic and novel two-stage strategies in sensitivity analysis 1. Fig D. The curves of pooled average and province-specific BD-temperature ERRs in sensitivity analysis 2. Fig E. The pooled average and province-specific ERRs in the overall temperature ranges in sensitivity analysis 2. Fig F. The comparison of ERRs between the classic and novel two-stage strategies in sensitivity analysis 2. Fig G. The curves of pooled average and province-specific BD-temperature ERRs in sensitivity analysis 3. Fig H. The pooled average and province-specific ERRs in the overall temperature ranges in sensitivity analysis 3. Fig I. The comparison of ERRs between the classic and novel two-stage strategies in sensitivity analysis 3. Fig J. The curves of pooled average and province-specific BD-temperature ERRs in sensitivity analysis 4. Fig K. The pooled average and province-specific ERRs in the overall temperature ranges in sensitivity analysis 4. Fig L. The comparison of ERRs between the classic and novel two-stage strategies in sensitivity analysis 4. Fig M. The curves of pooled average and province-specific BD-temperature ERRs in sensitivity analysis 5. Fig N. The pooled average and province-specific ERRs in the overall temperature ranges in sensitivity analysis 5. Fig O. The pooled average and province-specific ERRs on the original scale within the overall temperature range using the TS-based and B-spline-based strategies. Fig P. Comparison of ERRs on the original scale between the TS-based and B-spline-based strategies in a multi-region study with slightly different exposure ranges (degree of the polynomial was set as 2). Fig Q. Comparison of ERRs on the original scale between the TS-based and B-spline-based strategies in a multi-region study with slightly different exposure ranges (degree of the polynomial was set as 3). Fig R. Comparison of ERRs on the original scale between the TS-based and natural-spline-based strategies in a multi-region study with slightly different exposure ranges. Text A. Province-level predictors in meta-regression. (DOCX)

## Acknowledgments

We thank Bullet Edits Limited for the linguistic editing and proofreading of the manuscript.

## Author Contributions

**Conceptualization:** Wei Wang, Bo Zhou, Fang Liao.

**Data curation:** Wei Wang, Yunqiong Wang, Lin Chen.

**Formal analysis:** Wei Wang, Yunqiong Wang, Fang Liao.

**Funding acquisition:** Bo Zhou.

**Investigation:** Yunqiong Wang, Lin Chen.

**Methodology:** Wei Wang.

**Project administration:** Fang Liao.

**Resources:** Wei Wang, Fang Liao.

**Software:** Wei Wang, Fang Liao.

**Supervision:** Bo Zhou, Fang Liao.

**Validation:** Yunqiong Wang, Lin Chen.

**Visualization:** Wei Wang.

**Writing – original draft:** Wei Wang, Yunqiong Wang.

**Writing – review & editing:** Wei Wang, Yunqiong Wang, Lin Chen, Bo Zhou, Fang Liao.

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
