## [Decision Letter · Decision Letter 0]

19 Aug 2023

Dear Dr Liao,

Thank you very much for submitting your manuscript "Inverted U-shaped association between bacillary dysentery and temperature: a new finding using a novel two-stage strategy in multi-region studies" for consideration at PLOS Neglected Tropical Diseases. As with all papers reviewed by the journal, your manuscript was reviewed by members of the editorial board and by several independent reviewers. In light of the reviews (below this email), we would like to invite the resubmission of a significantly-revised version that takes into account the reviewers' comments. 

The Authors are expected to address all the criticisms by all Reviewers. In particular, please assess the impact of spatial confounders and spatial heterogeneity on the results (Reviewers #1 and #3), provide justification for the extrapolation beyond the local temperature range and select the best model (Reviewer #2). In additional to these comments, please address:

1. Please specify clearly the confounders in the Methods and provide the rationale

2. Data should be described in the Methods

3. Extrapolation was carried out sometimes >20 degrees outside the normal monthly average over 14 years. This is outside the expected range even with extreme weather.

4. Figure 5 shows a general pattern with peak at high temperature, however sometimes the extrapolated RR went over 5 which is higher than all predicted RR (e.g. Yunnan, Xizang) within the realized temperature range

5. Different models were fitted and showed different results. The authors also present the data and fitting to allow further assessment.

6. Figure 6, the estimates based on B-spline usually peaked earlier than that based on TS-spline. Which one is more reliable?

7. Figure 7 shows a U-shape relationship. Consider the temperature range, the results for low temperature is very different from Figure 5 & 6. How to reconcile these results?

8. Further, lowest risk was achieved at around 20 degrees (Figure 7), while moderate to high risk was achieved at the same temperature in Figure 5 & 6. How to reconcile these results?

We cannot make any decision about publication until we have seen the revised manuscript and your response to the reviewers' comments. Your revised manuscript is also likely to be sent to reviewers for further evaluation.

Sincerely,

Eric HY Lau, Ph.D.

Academic Editor

Mathieu Picardeau

Section Editor

The Authors are expected to address all the criticisms by all Reviewers. In particular, please assess the impact of spatial confounders and spatial heterogeneity on the results (Reviewers #1 and #3), provide justification for the extrapolation beyond the local temperature range and select the best model (Reviewer #2). In additional to these comments, please address:

1. Please specify clearly the confounders in the Methods and provide the rationale

2. Data should be described in the Methods

3. Extrapolation was carried out sometimes >20 degrees outside the normal monthly average over 14 years. This is outside the expected range even with extreme weather.

4. Figure 5 shows a general pattern with peak at high temperature, however sometimes the extrapolated RR went over 5 which is higher than all predicted RR (e.g. Yunnan, Xizang) within the realized temperature range

5. Different models were fitted and showed different results. The authors also present the data and fitting to allow further assessment.

6. Figure 6, the estimates based on B-spline usually peaked earlier than that based on TS-spline. Which one is more reliable?

7. Figure 7 shows a U-shape relationship. Consider the temperature range, the results for low temperature is very different from Figure 5 & 6. How to reconcile these results?

8. Further, lowest risk was achieved at around 20 degrees (Figure 7), while moderate to high risk was achieved at the same temperature in Figure 5 & 6. How to reconcile these results?

Reviewer's Responses to Questions

**Key Review Criteria Required for Acceptance?**

**Methods**

-Are the objectives of the study clearly articulated with a clear testable hypothesis stated?

-Is the study design appropriate to address the stated objectives?

-Is the population clearly described and appropriate for the hypothesis being tested?

-Is the sample size sufficient to ensure adequate power to address the hypothesis being tested?

-Were correct statistical analysis used to support conclusions?

-Are there concerns about ethical or regulatory requirements being met?

Reviewer #1: (No Response)

Reviewer #2: (No Response)

Reviewer #3: (No Response)

**Results**

-Does the analysis presented match the analysis plan?

-Are the results clearly and completely presented?

-Are the figures (Tables, Images) of sufficient quality for clarity?

Reviewer #1: (No Response)

Reviewer #2: (No Response)

Reviewer #3: (No Response)

**Conclusions**

-Are the conclusions supported by the data presented?

-Are the limitations of analysis clearly described?

-Do the authors discuss how these data can be helpful to advance our understanding of the topic under study?

-Is public health relevance addressed?

Reviewer #1: (No Response)

Reviewer #2: (No Response)

Reviewer #3: (No Response)

**Editorial and Data Presentation Modifications?**

Reviewer #1: (No Response)

Reviewer #2: (No Response)

Reviewer #3: (No Response)

**Summary and General Comments**

Reviewer #1: Temperature stands as a crucial factor influencing BD incidence. Within the study, a two-stage strategy was proposed to explore the correlation between monthly temperature and BD incidence concerning the original temperature data. The study revealed a U-inverted shape in the average temperature-BD association. The manuscript is well-written; nevertheless, certain key issues need addressing before publication. The following points should be considered:

While numerous previous studies have documented a non-linear association between BD and climate factors, it is essential to provide a detailed explanation of the contribution of this study.

Additionally, the confidence interval for the association between BD and temperature in high values appears considerably large, potentially challenging the validity of the U-inverted shape conclusion. The reviewer suggests investigating potential spatial confounding of impact factors and recommends exploring the spatial heterogeneity of their association, such as by climate zone.

For a more comprehensive discussion, it is advisable to include more comparisons between this study and other relevant papers in the manuscript. This will help contextualize and enrich the findings, contributing to the overall understanding of the subject matter.

Reviewer #2: This manuscript looks at association between bacillary dysentery and temperature, and disparities between provinces in China. While this is an interesting study, I have significant concerns with the paper’s methodology. I believe that getting these methods points nailed down is critical for publishing in a high visibility place like PLOS Neglected Tropical Diseases.

1. Line 165: the sentence is not complete. the matrix composed of 1-5 and 1 is also “what” for natural splines?

2. Figure 1: please provide more information on the b1-b5 in the left panel and b1-b6 in the right panel. What does the term “b6” stand for?

3. Method: 

(1) The term "truncated polynomial splines" is confusing. As far as I understand, both B-splines and natural cubic splines are types of polynomial splines. What distinguishes proposed TS in the present study and other polynomial splines like B-spline and natural cubic spline?

(2) It is suggested to use smoothing splines rather than polynomial regression for the following reasons: First, smoothing splines yield a piecewise continuous function composed of many polynomials to model the dataset, while polynomial regression provides a single polynomial that models the entire dataset. Second, polynomial regression may not handle thresholds well, especially when certain values of the predictor strongly influence the response at a different value of the predictor (Nonlocal Behavior in Polynomial Regressions). However, thresholds are quite common in modeling the effect of temperature on health. Third, smoothing splines are better suited for extrapolation, even at extreme values. Has the proposed method addressed or resolved these issues? To demonstrate the superiority of the novel method over classic methods, the authors should provide additional tests and evidence supporting its better performance. 

(3) The authors should refrain from assessing the results of B-splines based on "common epidemiological sense." The same knot setting or degrees of freedom may yield different curves and it is not uncommon for b-spline with three knots (10, 50, 90) to generate linear. For example, Liang et al. (https://doi.org/10.1186/s12940-016-0166-4) applied a cubic b-spline with 5 df for the daily mean temperature and 4 df in the lag space and results represented different patterns of temperature-health effects. For Figure 13, statistical tests should be provided to determine which model offers the best fit. 

(4) It is not necessary for the knots and bounds to be identical to generate basis variables with the same mathematical meaning among regions. Researchers can build region-specific basis variables, and likelihood-ratio tests can be employed to select the best model for each region or province in the current study.

(5) B-spline and natural cubic spline could also provide estimates on the original temperature scale. Can B-splines and natural cubic splines be truncated like what the authors did with polynomial regression? It would be helpful and meaningful to explore this possibility.

(6) Why was sunshine duration included in the model? Could it explain part of the effects of temperature?

4. Figure 6: can the authors provide justification for the extrapolation of the curve to temperatures beyond the local temperature range?

5. Figure S14-15 are not referenced in the main text.

6. Lines 275-276: The long-term trend should be characterized by a natural cubic spline with a df of 6 "per year." The phrase "per year" is missing in this statement.

Reviewer #3: 1. Spline seems too strict to fit the volatile real data;

2. Create a map of the disease and overlay it with relevant variates;

3. Visualize the data spatiotemporally; 

4. Measure and attribute (spatial) stratified heterogeneity, interpret the findings in epidemiology.

PLOS authors have the option to publish the peer review history of their article (what does this mean?). If published, this will include your full peer review and any attached files.

Reviewer #1: No

Reviewer #2: No

Reviewer #3: No
---

## [Decision Letter · Decision Letter 1]

24 Oct 2023

Dear Dr Liao,

Thank you very much for submitting your manuscript "Inverted U-shaped association between bacillary dysentery and temperature: a new finding using a novel two-stage strategy in multi-region studies" for consideration at PLOS Neglected Tropical Diseases. As with all papers reviewed by the journal, your manuscript was reviewed by members of the editorial board and by several independent reviewers. The reviewers appreciated the attention to an important topic. Based on the reviews, we are likely to accept this manuscript for publication, providing that you modify the manuscript according to the review recommendations. 

The authors have addressed all the comments by Reviewer #1 and #2. Reviewer #3 considered that the comments were not addressed, but after my further review I consider that the authors have addressed in some way though it may not be in the reviewer’s expected direction. Overall, I think the Authors have improved the manuscript substantially, and have a few minor comments:

1. An assessment of effect modification was made in Table 1. Please describe the methods and rationale in the Methods section.

2. Table 1 and R1, “beds” do you mean hospital beds?

3. L199, 213, “avoiding the trouble of the singular matrix”. Please use another word for “trouble”

Sincerely,

Eric HY Lau, Ph.D.

Academic Editor

Mathieu Picardeau

Section Editor

The authors have addressed all the comments by Reviewer #1 and #2. Reviewer #3 considered that the comments were not addressed, but after my further review I consider that the authors have addressed in some way though it may not be in the reviewer’s expected direction. Overall, I think the Authors have improved the manuscript substantially, and have a few minor comments:

1. An assessment of effect modification was made in Table 1. Please describe the methods and rationale in the Methods section.

2. Table 1 and R1, “beds” do you mean hospital beds?

3. L199, 213, “avoiding the trouble of the singular matrix”. Please use another word for “trouble”

Reviewer's Responses to Questions

**Key Review Criteria Required for Acceptance?**

**Methods**

-Are the objectives of the study clearly articulated with a clear testable hypothesis stated?

-Is the study design appropriate to address the stated objectives?

-Is the population clearly described and appropriate for the hypothesis being tested?

-Is the sample size sufficient to ensure adequate power to address the hypothesis being tested?

-Were correct statistical analysis used to support conclusions?

-Are there concerns about ethical or regulatory requirements being met?

Reviewer #2: (No Response)

Reviewer #3: my concerns numbered 1 and 4 were not appropriately addressed, such the problems remain unsolved.

**Results**

-Does the analysis presented match the analysis plan?

-Are the results clearly and completely presented?

-Are the figures (Tables, Images) of sufficient quality for clarity?

Reviewer #2: (No Response)

Reviewer #3: my concerns numbered 1 and 4 were not appropriately addressed, such the problems remain unsolved.

**Conclusions**

-Are the conclusions supported by the data presented?

-Are the limitations of analysis clearly described?

-Do the authors discuss how these data can be helpful to advance our understanding of the topic under study?

-Is public health relevance addressed?

Reviewer #2: (No Response)

Reviewer #3: my concerns numbered 1 and 4 were not appropriately addressed, such the problems remain unsolved.

**Editorial and Data Presentation Modifications?**

Reviewer #2: (No Response)

Reviewer #3: my concerns numbered 1 and 4 were not appropriately addressed, such the problems remain unsolved.

**Summary and General Comments**

Reviewer #2: Authors have adequately addressed my comments. I have no further comments.

Reviewer #3: my concerns numbered 1 and 4 were not appropriately addressed, such the problems remain unsolved.

PLOS authors have the option to publish the peer review history of their article (what does this mean?). If published, this will include your full peer review and any attached files.

Reviewer #2: No

Reviewer #3: No

Figure Files:

Data Requirements:

Reproducibility:

References

---

## [Editor Report · Decision Letter 2]

5 Nov 2023

Dear Dr Liao,

We are pleased to inform you that your manuscript 'Inverted U-shaped association between bacillary dysentery and temperature: a new finding using a novel two-stage strategy in multi-region studies' has been provisionally accepted for publication in PLOS Neglected Tropical Diseases.

Best regards,

Eric HY Lau, Ph.D.

Academic Editor

Mathieu Picardeau

Section Editor

Well done and congratulations on the excellent work!

---

## [Editor Report · Acceptance letter]

13 Nov 2023

Dear Dr Liao,

We are delighted to inform you that your manuscript, "Inverted U-shaped association between bacillary dysentery and temperature: a new finding using a novel two-stage strategy in multi-region studies," has been formally accepted for publication in PLOS Neglected Tropical Diseases.

Best regards,

Shaden Kamhawi

co-Editor-in-Chief

Paul Brindley

co-Editor-in-Chief
